# Short-Term and Long-Term Carcinogenic Effects of Food Contaminants (4-Hydroxynonenal and Pesticides) on Colorectal Human Cells: Involvement of Genotoxic and Non-Genomic Mechanisms

**DOI:** 10.3390/cancers13174337

**Published:** 2021-08-27

**Authors:** Liana C. Arnaud, Thierry Gauthier, Augustin Le Naour, Saleha Hashim, Nathalie Naud, Jerry W. Shay, Fabrice H. Pierre, Elisa Boutet-Robinet, Laurence Huc

**Affiliations:** 1Toxalim (Research Centre in Food Toxicology), University of Toulouse, INRAE, ENVT, INP-Purpan, UPS, 31027 Toulouse, France; Liana.Arnaud@inrae.fr (L.C.A.); Thierry.Gauthier@inrae.fr (T.G.); augustin.le-naour@inrae.fr (A.L.N.); saleha.hashim@inrae.fr (S.H.); nathalie.naud.31@inrae.fr (N.N.); Fabrice.Pierre@inrae.fr (F.H.P.); elisa.boutet@univ-tlse3.fr (E.B.-R.); 2Southwestern Medical Center Dallas, Department of Cell Biology, The University of Texas, Dallas, TX 75390, USA; Jerry.Shay@UTSouthwestern.edu

**Keywords:** colorectal carcinogenesis, pesticides, 4-hydroxynonenal, genotoxicity, non genomic carcinogenesis, genetic susceptibility

## Abstract

**Simple Summary:**

One’s environment, including diet, play a major role in the occurrence and the development of colorectal cancer (CRC). In this study, we are interested in two western diet associated food contaminants: 4-hydroxynonenal (HNE), a major lipid peroxidation product neoformed during digestion, and a mixture of pesticides to which we are commonly exposed to via fruit and vegetable consumption. The aim of this study was to analyse the impact of acute and long-term exposure to these contaminants, alone or in combination, on colorectal carcinogenesis. We used in vitro models of human colonic cells, either exhibiting or not different genetic susceptibilities to CRC. After acute exposure, we did not observe major alteration. However, long-term exposure to contaminants induce malignant transformation with different cellular mechanisms, depending on genetic susceptibility and contaminants alone or in mixtures.

**Abstract:**

To investigate environmental impacts upon colorectal carcinogenesis (CRC) by diet, we assessed two western diet food contaminants: 4-hydroxynonenal (HNE), a major lipid peroxidation product neoformed during digestion, and a mixture of pesticides. We used human colonic cell lines ectopically eliciting varied genetic susceptibilities to CRC: the non-transformed human epithelial colonic cells (HCECs) and their five isogenic cell lines with the loss of *APC* (Adenomatous polyposis coli) and *TP53* (Tumor protein 53) and/or ectopic expression of mutated *KRAS* (Kristen-ras). These cell lines have been exposed for either for a short time (2–24 h) or for a long period (3 weeks) to 1 µM HNE and/or 10 µM pesticides. After acute exposure, we did not observe any cytotoxicity or major DNA damage. However, long-term exposure to pesticides alone and in mixture with HNE induced clonogenic transformation in normal HCECs, as well as in cells representing later stages of carcinogenesis. It was associated with genotoxic and non-genomic mechanisms (cell growth, metabolic reprogramming, cell mobility and epithelial-mesenchymal transition) depending on genetic susceptibility. This study demonstrated a potential initiating and promoting effect of food contaminants on CRC after long-term exposure. It supports that these contaminants can accelerate carcinogenesis when mutations in oncogenes or tumor suppressor genes occur.

## 1. Introduction

Colorectal cancer is a public health issue that has affected 1.9 million people worldwide, of both sexes and all ages, in 2020 [1]. In terms of incidence, it is the second most common cancer in women and the third most common in men [1]. The vast majority of CRC cases are sporadic, and only 5–10% are attributable to inherited mutations associated with familial cancer syndromes [2]. CRC is a multistep progressive disease. Genetic factors are important, with a high frequency of mutation of the *APC* (*Adenomatous polyposis coli*) gene. This mutation is the initiating step of CRC occurring in 70–80% cases. Subsequent genetic alterations contribute to tumorigenesis such as the activation of proto-oncogene *K-RAS* (Kristen-ras) (40% of CRC) and the loss of the tumour suppressor *TP53* (Tumor protein 53) (70% of CRC) [3]. APC is involved in the regulation of the WNT signalling pathway and is a negative regulator of beta-catenin activation, important in the proliferation of stem cells in the bottom of the colonic crypt. K-RAS mediates proliferation induced by growth factors and TP53 is involved in the DNA repair, cell cycle and apoptosis induction.

However, the pattern of CRC incidence worldwide is heterogenic and associated with the westernization of dietary and lifestyle habits and with demographic ageing [1,4]. Physical inactivity, obesity, smoking and dietary factors (processed and red meat, alcohol) increase the risk of CRC [5]. Epidemiological studies in the UK indicate that 30 to 70% of all CRC cases are attributable to diet [6]. A western-style diet, rich in saturated fats of animal origin such as red meat, and poor in fibres, is recognized as one of the main risk factors for CRC [7,8]. 4-hydroxynonenal (HNE), is a major product of lipid peroxidation. HNE formation in the body is increased after food intake containing, for example, haem iron (from red meat) and omega 6 fatty acids. HNE is able to induce adducts in proteins and DNA and is involved in several cellular functions including cell proliferation, cell survival, differentiation, autophagy, senescence and apoptosis [9]. HNE acts as a selective pressure that favours the survival of preneoplastic cells over normal cells and contributes to the promotion of colorectal carcinogenesis [10].

Moreover, related to the extended use of pesticides, consumers are exposed chronically to a large number of food contaminants such pesticides that can affect health. The European Food Safety Authority (EFSA) report, which analysed the residues of pesticides in European food samples (fruits and vegetables), showed in 2017 that 54% of food tested was below the maximal residue limits and 27.5% contained two or more pesticide residues. The maximum number of residues in a single sample (peppers) was 30. There is a large number pesticide mixtures, therefore it is very complex to evaluate their combined effects [11]. In this work, we chose a representative dietary mixture of pesticides to which the French population was the most exposed to in food in 2006, based on a study carried out by the French Agency for Food, Environmental and Occupational Health & Safety (ANSES). [12,13]. The mixture is composed of five pesticides in the same proportions as French dietary exposure: four fungicides (Procymidone, Iprodione, Cyprodinil and Fludioxonil) and one insecticide (Lambda-Cyhalothrin). Procymidone and Iprodione are dicarboximide fungicides. When assessed in HepG2 cells, the cytotoxicity of the mixture seemed to rely on the toxicity of fludioxonil at 10 and 30 µM only. The detection of total γH2AX to evaluated DNA damage showed that fludioxonil (6 µM) and cyprodinil (20 µM) alone gave a positive signal. When pesticides are mixed at equimolar concentrations, genotoxicity occurs, whereas none of them are genotoxic individually [12,14].

Dicarboximide fungicides are known to induce lipid peroxidation in fungi [15], mammalian cells [16], primary cultured trout hepatocytes [17] and impact the human respiratory system in vitro [18]. Procymidone is an endocrine disruptor [19] and it is known to bind the androgen receptor (AR) and acts as an AR antagonist in vivo and in vitro, inhibiting androgen-dependent gene expression by inhibiting AR-binding to DNA [20]. Iprodione is a steroidogenesis inhibitor [19]. Cyprodinil is a pyrimidiamine fungicide that induces the phosphorylation of the extra cellular signal-regulated kinase (ERK) that phosphorylates growth and transcription factors and regulates proliferation, differentiation, survival and migration [21] in mammalian cells and also activated ER (estrogen receptor) signalling [22]. Cyprodinil is an activator of the Aryl hydrocarbon receptor (AhR) and induces Ah target genes such as cytochrome P450 (CYP) 1A1 in ovarian granulosa cells [23,24]. Fludioxonil is a phenylpyrrole fungicide that is a potential activator of AhR, AR and ER [24]. Cyprodinil and Fludioxonil are able to induce neurotoxicity in human neuronal and glial cell lines in vitro. Lambda-Cyhalothrin is a synthetic pyrethroid (SP) [25]. Several studies show that SPs represent a class of endocrine disrupting chemicals (EDCs) inducing a dysregulation of biosynthesis, metabolism or action of hormones [26,27]. Furthermore, a childhood exposure to pyrethroid insecticides is associated with a decrease of neurocognitive abilities in children at six years [28].

However, the scientific literature does not cover the areas of multiple pesticide exposure and health consequences. In vivo studies on mixtures of pesticides have shown that dietary exposure may promote the occurrence of chronic diseases, but no mechanistic results are available to date [29,30]. The occurrence of CRC results from the interplay of genetic, environmental and microenvironmental factors [31]. In this study, we investigated if genetic alterations modify cellular responses to food contaminants. For this purpose, we used isogenic human epithelial colonic cells (HCECs) exposed to an original and never tested combination of food contaminants, HNE and/or the mixture of pesticides. These cells are non-malignant. They were immortalized by cyclin-dependent kinase 4 (Cdk4) and the human ribonucleoprotein enzyme telomerase (hTERT). HCEC have no multiple cytogenetic changes [32]. By this process, HCECs do not acquire tumorigenic properties and have a normal karyotype. Five cell lines have been experimentally obtained from the non-transformed HCECs (CT cells): three of them mimic the driver mutation genes found in CRC: loss of *APC* (CTA cells) and *TP53* (CTP cells) and ectopic expression of mutant *KRAS* (CTR cells) [3]. One contains and expresses the three driver mutations (CTRPA cells) and another one has these three alterations and the ectopic expression of the 1309-aa truncated form of APC (CTRPAt cells), the most frequently found form in human sporadic CRC. The six cell lines mimic the different steps of CRC, initiation (CT cells), promotion (CTA, CTR and CTP cells) and progression (CTRPA and CTRPAt cells) and constitute an integrative tool to study the impact of chemicals according to the stages of human colorectal carcinogenesis and genetic predisposition [33].

Previous studies separately analysed the impact of HNE [10] and this mixture of pesticides [13] at high concentrations (HNE 40 µM and pesticides 100 µM). Here we analysed the impacts of long-term exposure to lower concentrations of HNE (0.5 µM) and to a mixture of pesticides (10 µM) on these six cell lines and analysed the acquisition of cancer hallmarks potential. We conducted an original mechanistic toxicological study on the effects of a mixture of pesticides and a neoformed compound during digestion in colorectal carcinogenesis. This study allows a better understanding of the impact of environment through a complex mixture of food contaminants relative to genetic susceptibility to CRC.

## 2. Materials and Methods

### 2.1. Chemicals and Supplements for Cell-Culture-Media

The 4-hydroxynonenal (HNE) was provided by Clinisciences (Nanterre, France). Fetal Bovine Serum (FBS) provided by ThermoFisher (Eindhoven, Netherlands). Puromycin, hydromycin and zeocin were provided by Fisher Scientific(Hampton, NH, USA). Epidermal growth factor (EGF), hydrocortisone, insulin, transferrin, sodium selenite (5 nM) and gentamycin sulfate (5 µg/mL), DMSO (dimethylsulfoxyde), procymidone, iprodione, cyprodinil, fludioxonil, lambda-cyhalothrin, methyl methanesulfonate (MMS), Trifluoromethoxy carbonylcyanide phenylhydrazone (FCCP), rotenone, oligomycin and etoposide were provided by Sigma-Aldrich (Saint-Quentin Fallavier, France). Ro 19-8022 was a gift from Hoffman Laroche Ltd. (Basel, Switzerland), and formamidopyrimidine-DNA glycosylase (Fpg) a gift from Serge Boiteux, CNRS, France.

### 2.2. Antibodies

Anti 53BP1 (Novus Biological, Abingdon, United Kingdom, ref: NB100-304) from rabbit was diluted 1/2000 in PBS containing 3% bovine serum albumin (BSA) and 0.1% Triton X-100 for the detection of 53BP1 nuclear foci. For the detection of γH2AX nuclear foci, anti γH2AX (Merck Millipore, Fontenay-sous-Bois, France, ref: 05-636) from mouse was diluted 1/2000 in PBS containing 3% BSA and 0.1% Triton X-100.

Anti-Paxillin (rabbit monoclonal, abcam, ref: ab32084), Phalloidin (for actin staining) (Sigma-Aldrich, ref: P1951), were diluted 1/250 and 1/2500 respectively in PBS containing 3% BSA and 0.1% Triton X-100.

Alexa fluor 568 (spectrum 578/603) from rabbit and 488 (spectrum 490/525) from mouse (ThermoFisher, Eindhoven, Netherlands, ref: A-11011 and A-11001) were diluted 1/800 in PBS containing 3% BSA and 0.1% Triton X-100.

E-Cadherin (1/200), Vimentin (1/1000) and Actin (1/10,000) proteins was detected by mouse monoclonal antibody (Abcam, Paris, France, ref: ab1416), rabbit monoclonal antibody (Abcam, Paris, France, ref: ab92547) and mouse monoclonal antibody respectively (Sigma-Aldrich, ref: A5441). Donkey anti-Rabbit IgG (H + L), HRP and Donkey anti-Mouse IgG (H + L) (ThermoFisher, Eindhoven, Netherlands, ref: SA1-200 and SA1-100 respectively) was used as secondary antibody. The secondary antibodies were diluted 1/1000 in TBS/BSA 3%/0.1% Tween.

### 2.3. Cell Lines and Maintenance

Isogenic human colorectal cell lines (HCECs), generated and provided by Pr Jerry W Shay [32], were maintained on Primaria^TM^ flasks (Dominique Dutscher, Bernolsheim, France) in a humidified atmosphere with 5% CO_2_ at 37 °C, in 4:1 high-glucose Dulbecco modified Eagle medium/medium 199 supplemented with 2% FBS, epidermal growth factor (EGF 20 ng/mL), hydrocortisone (1 mg/mL), insulin (10 mg/mL), transferrin (2 mg/mL), sodium selenite (5 nM) and gentamycin sulfate (50 µg/mL). In addition, CTA cells were selected by puromycin (1µg/mL), CTR cells by hygromycin (200 µg/mL) and CTP cells by zeocin (1 mg/mL). Malassez Counting Chamber (Dominique Dutscher, Bernolsheim, France) was used for study the growth of cells.

### 2.4. Treatments

Cells are exposed to DMSO (1/2000) as control, etoposide (25 nM or 10 µM), MMS or Ro-19-8022 as positive controls of genotoxicity, HNE (0.5 µM in water), pesticides (the total concentration of mixture was 10 µM in DMSO) with the mixture composed of procymidone, iprodione, cyprodinil, fludioxonil and lambda-cyhalothrin (Table 1). The cells are exposed up to 24 h for the acute exposure or daily during 3 weeks with a splitting per week for the long-term exposure.

### 2.5. Viability Assay

Cell viability was first determined using the CellTiter-Glo^®^ Luminescent Cell Viability Assay (Promega) (Madison, WI, USA) according to the manufacturer’s instructions. Cell Titer-Glo^®^ determine the number of viable cells based on quantification of ATP level, which signals the presence of metabolically active cells. HCECs were grown on Primaria™ 96 well plates (8000 cells/well) during 24 h. Then, cells were exposed with the different treatments. Luminescence was read using Infinite 200 PRO reader (TECAN).

The XCelligence^®^ system (Ozyme, St Cyr L’Ecole, France) was used according to the manufacturer’s instructions (Ozyme). HCECs were seeded on electronic microtiter plates (E-Plate) with 8000 cells/well. After 24 h, when the cell index (CI) was stable, cells were treated with the different treatments. Cell impedance was measured in each well every 1 h for 24 h. Impedance signals were analyzed by an integrated software RTCA Analyzer Version 2.0 (Ozyme, St Cyr L’Ecole, France) and expressed as a CI-value that reflects cell number, cell adhesion and/or cell morphology. Experiments were carried out independently in triplicate.

### 2.6. Immunofluorescence

Cells were grown on glass slide in 12-well plates (5000 cells/well) for 48 h. After 24 h or 3 weeks of food contaminant treatment, cells were washed with PBS and fixed with 4% of paraformaldehyde for 20 min on ice. Cells were washed with PBS for 5 min then permeabilized with 0.5% Triton X-100 in PBS for 15 min. Cells were blocked with 3% BSA in PBS for 1 h and incubated with primary antibodies anti 53 BP1 (rabbit) and anti γH2AX (mouse) or anti paxillin (rabbit) and anti-phalloidin for 2 h. After three washes, cells were incubated with secondary antibodies Alexa fluor 568 and Alexa fluor 488 for 1 h and nuclei were labelled with 4,6-diamino-2-phenyl indole (DAPI). Coverslips were mounted with Prolong Gold. Slides were analysed with a confocal laser-scanning microscope (SP8, LEICA, Nanterre, France) equipped with a 40× oil immersion objective and using 405, 488 and 565 nm lasers to reveal DAPI, Alexa 488 and Alexa 568, respectively. For the quantification of DNA damage, the images were analysed using ImageJ software and cells were scored positive when containing more than five γH2AX or 53BP1 foci. For the mobile cell parameters (anchoring points, lamellipodia, filipodia, stress fibers, pseudopods and cortical actin), the analysis of fluorescent confocal images (z-stack, µM-sized step) was done by blinded visual scoring by two independent experimenters. For this reason, we considered these parameters as semi-quantitative. No statistical analyses were performed on these results and they were described only as a range of effects.

### 2.7. Comet Assay

The alkaline comet assay was used to detect strand breaks and alkali-labile sites as previously described [34]. To induce DNA damage as products of purine oxidation detectable by formamidopyrimidine-DNA glycosylase (Fpg), defrosted human peripheral blood mononuclear cells (PBMC) (120，000 cells/mL) were placed on ice and treated with the compound Ro 19–8022 (at 1 mM in PBS) during 2 min 30 s under visible light (1000 W-halogen). Ro 19–8022 plus visible light exposure is an appropriate positive control for the Fpg-modified comet assay [35]. Cells were then pelleted for 10 min at 200× *g* at 4 °C. Briefly, trypsinized HCECs were embedded in 0.7% low melting point agarose and deposit on Gelbond^®^ (Sigma-Aldrich, Saint-Quentin Fallavier, France) prior to lysis and electrophoresis. A parallel digestion with Fpg enzyme allowing the detection of Fpg-sensitive sites was performed, as described [34]. Fifty cells per deposit and two deposits per sample were analysed. The extent of DNA damage was evaluated for each cell by measuring the intensity of all tail pixels divided by the total intensity of all pixels in the head and tail of the comet. The median of these 100 values was calculated and named % tail DNA. The Net level of Fpg-sensitive sites (% Tail DNA) was obtained for each condition by subtracting the damage (% Tail DNA) obtained in the absence of Fpg from Fpg-exposed comets (“+Fpg” − “−Fpg” = Net Fpg). This Net Fpg level represents mainly oxidized bases. HCEC cells and PBMC were treated with methyl methanesulfonate (MMS) to induce DNA damage. MMS is a positive control to the experiment.

### 2.8. Western Blot Analysis

Cells were washed in ice-cold PBS, scrapped and pelleted by centrifugation. The whole lysates were collected in electrophoresis sample buffer containing 50 mM Tris-Base/150 mM NaCl (pH 7.5), 1% Triton X100, 2% sodium deoxycholate, and 2% sodium dodecyl sulfate. Lysates were further homogenized by sonication on ice and heated at 100 °C for 5 min. Protein concentrations were measured by DC™ Protein Assay (BIO-RAD, Marnes La Coquette, France). Proteins (10 µg/well) were separated by SDS-PAGE and blots were transferred to nitrocellulose membranes. Membranes were then saturated and incubated with primary antibodies E-cadherin and Vimentin at 4 °C for a night under agitation. Secondary antibodies conjugated to horseradish peroxidase (HRP) enzyme were incubated for 30 min and membranes were revealed using Pierce™ Western Blot Signal Enhancer (ThermoFisher, Eindhoven, Netherlands, ref: 21050) and ChemiDoc™ Touch Imaging System (BIO-RAD, Marnes La Coquette, France). The quantification was realised with Image J software. (Ver. Image J 1 .52a, accessed on 05 July 2021)

### 2.9. Soft Agar Clonogenicity Assay

In a 6-wells plate coated with a lower layer media containing Noble agar 0.5%, cells were seeded in medium 0.375% Noble agar at 5000 cells per well, in triplicate. After 10 days, colonies were counted in the whole well. Experiments were performed in three independent experiments.

### 2.10. Seahorse

Cells were grown on Seahorse 96-well plates (24,000 cells/well) for 24 h. Cells were washed and incubated in base medium (Agilent Technologies, Les Ulis, France) at 37 °C for 1 h. Extracellular Acidification Rate (ECAR) and Oxygen Consumption Rate (OCR) were measured in real-time with Glycolysis Stress Test Kit and Mito Stress Test Kit respectively using the Seahorse XFe96 Analyser (Agilent Technologies, Les Ulis, France) following manufacturer’s instructions. Data were normalized by protein content that was measured by the DC™ Protein Assay (BIO-RAD, Marnes La Coquette, France).

### 2.11. Statistical Analysis

For gaussian parameters (continue responses with normal distribution), a two-way analysis of the variance is carried out with contaminants, cell lines and interaction as fixed factors. A Log transformation is applied when Log-Normal distribution is detected on the parameter. Parameters resulting from a counting are fitted with a generalized linear model assuming a negative binomial distribution. Post-hoc tests for pairwise comparisons on contaminants or cell lines are provided without adjusted for multiplicity to preserve the beta-risk level in the context of the study (safety purpose). Statistical significance is set at 5% alpha-risk level (*p*-value ≤ 0.05). Descriptive summary and graphical representations are provided by contaminants and cell lines using arithmetic mean and standard deviation of the mean on actual values.

The pairwise differences of cell lines at the DMSO level are performed on the observed values. For the other analyses, comparisons of contaminants versus DMSO at a fixed level of the cell line and comparisons of cell lines at a fixed level of contaminant (DMSO excluded), the treatments are carried out on the values focused on the average of DMSO within the respective cell line. These data pre-processing normalizes the effect of contaminants as a variation from the average DMSO of each cell line.

The statistical analysis is summarized in tables in Appendix A.

As complementary analysis, a non-supervised Principal Component Analysis (PCA) was produced to describe the correlation structure of the carcinogenic events. Data analyses were performed using “R: A language and environment for statistical computing” (R Core Team (2016)) and SAS/STAT^®^ software 9.4 (SAS Institute Inc., Cary, NC, USA).

## 3. Results

For this study, we chose a combination of pesticides that includes four fungicides (Procymidone, Iprodione, Cyprodinil and Fludioxonil) and one insecticide (Lambda-Cyhalothrin) (Table 1). This pesticide combination (at 10 µM) was used alone or in mixture with 0.5 µM of HNE (HNE-Pest). Previous studies separately analysed the impact of HNE [10] and this mixture of pesticides [13] at much higher concentrations, 40 µM and 100 µM respectively. The six isogenic HCECs were exposed for 24 h (acute) and for three weeks (long-term treatment), to characterize toxicity and carcinogenic events.

### 3.1. Acute Viability and DNA Damage of Contaminants in HCECs

We firstly evaluated acute viability after 24 h exposure by measuring the cellular ATP concentration (Figure 1A). We observed differences of cellular ATP levels between cell lines independently of the treatments: CTA, CTP and CTRPA cells have higher level of cellular ATP compared to CT cells. For each cell line taken separately, treatments did not have significant effects (versus control). We also measured cellular impedance to follow in real time the cell index (CI) (Figure 1B). The arbitrary unit CI represents the number of cells, cellular adhesion and shape. The six isogenic HCECs exhibit different CI, reflecting the different morphologies and shapes related to the genetic modification. For each cell line taken separately, treatments did not have significant effects after 24 h. In conclusion, treatments did not affect ATP levels and cellular impedance. We just evidenced some difference between isogenic cells, irrespective of the treatments.

In absence of major impact in cell viability at these concentrations, we were able to assess acute DNA damage, such as DNA breaks and DNA oxidation. For this purpose, we performed alkaline comet assay to detect DNA single- and double-strand breaks.

We also carried out Fpg-comet assay to detect oxidative lesions of DNA, mainly 8-oxoguanine. The addition of FPG enzyme allows enzymatic conversion of 8-oxoguanine into DNA breaks, which is detected by alkaline comet assay. The Fpg-comet assay has been described to be a valid and sensitive marker of oxidative DNA lesion [36,37]. After 2 h of treatment with pesticides, comet assay revealed a significant increase in DNA breaks induced by pesticides in CT and CTA cells. HNE exposure had no effect irrespective of the cell line. The HNE-Pest treatment induced DNA breaks in CT, CTP, CTRPA cells, (Figure 1C). No oxidative DNA damage was induced by the treatments, as the addition of Fpg enzyme did not reveal any Fpg-sensitive sites. MMS and Ro 19-8022 were used as positive controls (Appendix A).

DNA damage has been also evaluated by immunofluorescence with antibodies directed against 53BP1, which is a biomarker of double-strand breaks (DSB). 53BP1 is recruited to DSB and forms nuclear foci at DNA damaged sites [38] (Figure 1D). We used etoposide as a positive control of DSB (Appendix A). HNE increased the number of foci in CT, and much less in CTRPA cells. Pesticide treatment increased the number of foci in CT and in CTA cells while HNE-Pest treatment increased the number of foci in CT, CTRPA and CTRPAt cells (Figure 1E). These results also showed that some cells had different sensitivity towards treatments (interaction cell line x treatment *p* = 0.021, Appendix A). CT cells appeared more sensitive to all the treatments compared to the other cell types.

### 3.2. Cell Growth and Clonogenicity after a Long-Term Exposure to Contaminants in HCECs

We investigated the impact of the contaminants on the HCECs after a long-term exposure. The different cell lines were exposed to HNE, pesticides and HNE-Pest daily for three weeks.

These experiments aimed at characterizing the whole capacity of cells to proliferate and survive and to growth after splitting in the presence of chemicals. Then, the impact of long-term exposure on cell growth during three weeks was evaluated by counting cells at each splitting (each week). Here we present in Figure 2A, the last splitting after three weeks of exposure. The comparison of all the untreated cells showed that CTP and CTRPA cells exhibited a higher number of cells after three weeks. When considering the treatments, we observed no effect of HNE and pesticides alone, whereas the mixture HNE-Pest induced a significant increase in the number of cells in CTA, CTP and CTRPAt cells.

The capacity of the contaminants to induce malignant transformation (hallmark of cancer) in the six cell lines after a three-week exposure was analysed by the soft agar assay (Figure 2B). Non-malignant cells do not grow without anchoring and are not able to grow in soft agar. However, cancerous cells can acquire the ability to grow without anchoring and thus grow in soft agar. The exposure to pesticides and HNE-Pest induced the increase in the number of colonies in all isogenic cell lines. HNE induced a significant increase only in CTA, CTP and CTRPA cells. The CTA, CTP and CTRPA cells were the cell lines the more responsive to HNE. CTP and CTRPA cells are more responsive to pesticides and CTA cells were more responsive to HNE-Pest, in comparison with the others cell lines.

In the majority of the tested conditions, contaminants are able to transform non-mutated cells (CT cells) and genetically predisposed cells (CTA, CTR, CTP cells) and to increase the tumorigenesis in advanced-stage cells (CTRPA and CTRPAt cells).

### 3.3. Long-Term DNA Damage of Contaminants in HCECs

In an attempt to explain the capacity of the contaminants to induce malignant transformation in the different cell lines, DNA damage was analysed after a long-term exposure (3 weeks) to the contaminants by immunofluorescence using antibodies directed against γH2AX (phosphorylation of H2AX on Ser139) (Figure 3A) and 53BP1 (Figure 3B). They are both biomarkers that signal DNA damage or possibly non- homologous end-joining involving DSB [39]. A long-term exposure treatment with HNE, pesticides and the mixture HNE-Pest induced a significantly increased proportion of γH2AX positive cells in all the cell lines (Figure 3A).

In untreated cells, the proportion of 53BP1 positive cells was more important for CTP, CTR and CTRPA cells compared to CT cells. The pesticide treatment and the mixture HNE-Pest induced an increase in the proportion of 53BP1 positive cells in CT cells. After exposure to HNE-Pest, the proportion of 53BP1 positive cells was less for CTA, CTP and CTRPA cells compared CT cells. As 53BP1, detected in nuclear foci, is a more specific DSB signalling biomarker than γH2AX [40,41], this could explain the higher level a positive γH2AX-foci positive cells, which could correspond to signalling of DSB but also replicative stress, or other chromatin modifications.

### 3.4. Metabolic Reprogramming after Long-Term Exposure to Contaminants in HCECs

To evaluate the impact of contaminants on cell energy metabolism, Extracellular Acidification Rate (ECAR) and Oxygen Consumption Rate (OCR) were measured in real-time using Seahorse^®^ technology. Cells were submitted to a MitoStress protocol to challenge the mitochondrial function (Figure 3A).

We observed that HNE-Pest treatment induced an increase in basal OCR and basal ECAR (Figure 4B) in the cell lines altered or mutated for APC function (CTA, CTRPA and CTRPAt cells. Since this treatment also increased the maximal respiratory rate (Figure 4C), it could be assumed that the higher basal respiratory rate was due to a better efficiency of the electron transport chain. Thus, these cells were more sensitive to HNE-Pest and exhibited a higher metabolic level. Moreover, the capacity of ATP production of all cell lines except CT cells, reached a higher level when exposed to HNE-Pest (Figure 4D). This capacity can contribute to the cellular fitness supporting carcinogenic process triggered by the mixture HNE-Pest.

### 3.5. Cell Morphology after Long-Term Exposure to Contaminants in HCECs

In order to evaluate the effects of treatment on cell morphology, actin and paxillin were stained respectively with dye-labelled phalloidin and antibody (Figure 5). We considered the following parameters which are the hallmarks of moving cells: anchoring points, lamellipodia, filipodia, stress fibers, pseudopods and cortical actin. Cells showing at least one of these parameters were considered as moving cells.

In control conditions for the six cell lines, we observed that around 20% of cells were mobile, except for CTRPAt cells exhibiting a slight increase in cell mobility (around 30%). After long-term exposure, HNE had no effect on mobility parameters. The treatment by pesticides and HNE-Pest increased the number of mobile cells in all cell lines except in CTRPAt cells which may be due to their higher basal mobility. The maximal effect was around 60–75% of mobile cells.

### 3.6. Epithelial-Mesenchymal Transition (EMT) after Long-Term Exposure to Contaminants in HCECs

EMT can also be involved in tumoral transformation following contaminants exposure. We compared the effects of treatments in each cell line (Figure 6 and Appendix A) on the level of E-cadherin and vimentin, two markers of the EMT process. Pesticide and HNE-Pest treatment significantly decreased the level E-cadherin in CTA cells. We observed a tendency of decrease in CT and CTRPAt cells. There is no statistical difference between treatments per cell line regarding vimentin.

### 3.7. Principal Component Analysis of Long-Term Exposure Results to Contaminants in HCECs

To determine whether the examined carcinogenic events correlated, we carried out a principal component analysis (PCA) of long-term exposure results (Figure 7). On the two-dimensional projection here after which support about 62% of the total variability, we observe a correlation between cellular metabolism (including ATP production, basal OCR maximal OCR and ECAR), another correlation between the results of DNA damage signalling (γH2AX and 53BP1 immunostaining) and finally, a correlation between the results of EMT (E-cadherin and Vimentin immunoblots). However, no result was strongly correlated with the malignant transformation observed with soft agar assay. This suggests that the acquisition of the major carcinogenic transformation with growth without anchoring is not simply the result of single cellular alteration, but of a combination of several mechanisms, probably DNA damage signalling, cell growth and energetic metabolism.

## 4. Discussion

The link between cancer and the environment is a controversial subject that has been brought up to date these last years [42,43]. Using a mathematical correlation analysis, scientists estimated that two thirds of tumours are due to random chance rather than environmental or genetic factors. Wu et al. (2016) reassessed these data and concluded that extrinsic factors play a major role in the occurrence of cancer [44]. By joining this debate supported by philosophers of cancer biology [45], we questioned the role of environmental factors, especially food, in the onset of colorectal carcinogenesis. We analysed the impact of two types of food contaminants used alone or in combination: HNE, a neoformed compound, and a mixture of pesticides. HNE is produced during digestion of iron-containing food, such as red meat [10,46]. This process involves microbiota [47,48]. Moreover, we chose a representative mixture of pesticides to which French consumers are exposed daily [13]. We used this mixture of pesticides reflecting a real food exposure to carry out mechanistic studies relative to its carcinogenic potential. In order to study acute toxicity but also long-term exposure, more relevant for carcinogenesis studies, we carried out experiments after 24 h and three weeks of exposure. To study early and progressive steps of colorectal carcinogenesis, we use non transformed human epithelial colonic cells (HCECs) and their five isogenic cells lines [32]. By working with these six isogenic cell lines, we could compare the differences of susceptibility regarding contaminants and we hypothesized that contaminants can affect different stages of colon carcinogenesis.

We first studied classical toxicological endpoints, e.g., cell viability and genotoxicity, after acute (2 h or 24 h) and long-term (three weeks) treatments with the food contaminants (at 0.5 µM for HNE and 10 µM for pesticides). Previous studies demonstrated that the same mixture of pesticides was able to induce cytotoxicity on a colonic cell line (LS-174T) after 24 h at a concentration of 100 µM [13], 10 times higher than those used here. Other studies showed that HNE induced apoptosis at 40 µM [10,46], a concentration 80 times greater than the concentration used in our study. In contrast, using lower concentrations, we observed discrete effects after acute treatment: no modification of cell viability and some discrete DNA damage differences with pesticides and HNE-Pest, notably DNA breaks. For analysed DNA damage, we used two methods: alkaline comet assay (to detect both single and DSB) and 53BP1 foci (to detect the signalling of DSB), which are complementary. For example, in CT cells, pesticides and HNE-Pest induced a significant effect with both assays, suggesting induction of DSB (without excluding single-strand breaks). When only the comet assay is positive, as in CTP cells exposed to HNE-Pest, it suggests only single-strand breaks. Except for this discrete DNA damage, main differences were due to intrinsic genetic differences between cell lines. HNE, at 0.5 µM and 2 h exposure, barely induced 53BP1 foci in CT cells, whereas it was described to be genotoxic in cerebral endothelial cells for doses higher than 1 µM after 3 h of exposure [49]. A previous study proposed that low DNA damage induced by an acute exposure to a low concentration of environmental toxicant can lead to an efficient DNA damage response, i.e., an efficient DNA repair and maintenance of cellular integrity [50]. In contrast, long-term exposure to HNE seems to trigger γH2AX foci in all cell lines, suggesting that the maintenance of DNA integrity could not be longer assured and could contribute to mutagenesis, in particular G:C > T:A mutations in proto-oncogenes like *K-RAS* [51]. In these experimental conditions, in which cells were cultivated and exposed for three weeks, cellular aging is also a parameter that could influence basal DNA damage. However, the level of γH2AX foci in control cells was very low, underlying that aging was not major in the observation of the effects of the exposure.

When mimicking long-term exposure and studying a broader panel of toxicological endpoints, we identified important changes supporting carcinogenesis. Indeed, major changes in cell growth, morphology related to mobility, DNA damage signalling, growth without anchoring and cell metabolism for *APC*-modified cells were observed. Thus, our study strongly indicates that long-term exposure is a major condition that should be taken into account for real-life food contamination assessment. It is also supported by previous studies demonstrating different genotoxic and metabolic effects of pesticides after long-term exposure compared to acute exposure [52,53]. These results also highlight the fact that environmental carcinogenesis involves multiple mechanisms depending on the type of contaminant and genetic susceptibility [5]. In vivo experiments could confirm our mechanistic hypothesis: by injecting the transformed cells after long-term exposure into nude mice, we could evaluate their malignant potential. Alternatively, by feeding azoxymethane-induced rats with a diet containing haem iron and pesticides, we could evaluate the preneoplastic incidence after three months [46] and the involvement of gut microbiota [54]. To conciliate in vitro and in vivo approaches, in order to investigate carcinogenesis and the related mechanisms, zebrafish could constitute a promising model [55].

In order to study the carcinogenesis potency of the toxicants, we evaluated the capacity of contaminants to induce a step in malignant transformation after a long-term exposure (three weeks) using the Soft Agar assay. The Soft Agar assay allows the evaluation of the ability of cells to grow without anchorage, a hallmark of carcinogenic transformed cells. CT cells, considered as normal HCECs, acquired the capacity to grow without anchorage only after a long-term exposure to pesticides and HNE-Pest. Thus, we demonstrated for the first time that these contaminants could induce CRC initiation. Genetically- predisposed cells (CTA, CTR and CTP cells) and advanced stage cells (CTRPA and CTRAt cells) also acquired an increase in the capacity to grow without anchorage in the majority of conditions of exposure. This suggests that pesticides alone or in mixture with HNE could promote CRC and increase tumorigenesis in advanced stages. Moreover, we suggest that contaminants in food can enhance carcinogenesis when cells exhibit genetic susceptibility, as already mentioned by Bermejo & Hemminki [56].

In a previous study, we analysed the potential tumorigenic role of Cytolethal Distending Toxin (CDT) in HCECs [33]. In these conditions, the genotoxic CDT did not induce CRC initiation but favored CRC promotion. The induction of malignant transformation by CDT was also associated with DNA damage. However, in the present work, genotoxicity was not related to the malignant transformation observed after acute or long-term exposure to the contaminants, as supported by the PCA analysis. Therefore, we investigated non-genomic carcinogenic effects, which are known to disrupt different signalling pathways [57,58].

We conducted a comparative study by analysing HCECs CT cells and five isogenic lines mimicking the main stages of colorectal carcinogenesis. We observed a differential impact of contaminants when the cells are genetically modified. Pesticides and HNE-Pest favour malignant transformation in CT cells and it was also associated with an increase in cell mobility. Transformation was observed in CTA cells with all the contaminants. Compared to CT cells, this transformation is associated with cell mobility changes, but also included increase of cell growth and changes in cell metabolism. CTA cells are more sensitive to HNE transformation compared to CT cells. The difference of sensitivity of normal CT cells and CTA cells was previously described after a short exposure to a higher dose of HNE and fecal water for heme-containing diet fed rats [10,46]. For CTR cells, transformation was induced by pesticides and HNE-Pest and it was associated only with increase of cell mobility. The ectopic expression of KRAS does not appear to affect cell mechanisms studied here after a long-term exposure to contaminants. All the contaminants lead to a transformation in CTP cells that are more sensitive than CT cells. Only changes of cell mobility were observed as for CT cells, while the cell growth is strongly increase in CTP cells compared to CT cells. The inactivation of the tumour suppressor TP53 favour cell proliferation in CRC via its role in the regulation of cell cycle [59,60]. Mixture HNE-Pest induced distinct cellular impact in CTRPA and CTRPAt cells compared to CT cells, especially concerning cell metabolism. Globally, we observed that the cell lines altered or mutated for *APC* (CTA, CTRPA and CTRPAt cells) exhibit changes in cell metabolism. CRC is associated with activation of Wnt signalling because *APC* mutations appear in the majority of CRC. Previous studies reported potential links between Wnt signalling and cancer metabolism in several cancer types [61,62,63,64]. It is important to note that the increase in cell mobility observed in some cell lines did not associate with epithelial-mesenchymal transition (EMT). EMT did not appear to be induced by the mixture of pesticides and/or HNE.

## 5. Conclusions

This works highlights the toxicity of food contaminants alone or in mixture on different isogenic cell human colonic epithelial cell lines with different genetic susceptibilities after acute and long-term exposure. Our results show that the carcinogenic effect of HNE and pesticides cannot be explained only by genomic alterations but by a combination of cell disturbances relative to cell growth, energetic metabolism and cell mobility. The procarcinogenic effect of foods is often analysed in terms of nutritional composition (meat, fibre, fat). However, our results show the need to take into account the level of contamination by pesticides, which constitute a chemical microenvironment that should no longer be ignored, in order to be able to conduct effective colorectal cancer prevention policies.

## Figures and Tables

**Figure 1 cancers-13-04337-f001:**
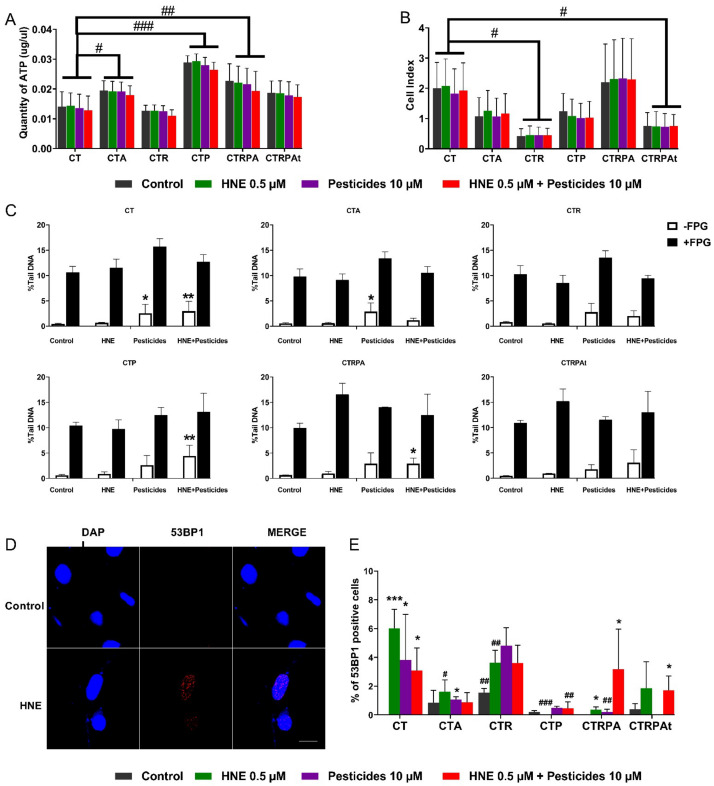
Acute viability and DNA damage induced by contaminants in HCECs. (**A**) Quantification of ATP levels at 24 h post-treatment using Cell Titer Glo^®^. (**B**) Cell index values (Xcelligence, as described in Materials and Methods) at 24 h post-treatment. (**C**) DNA damage evaluated by alkaline (-Fpg) and Fpg modified comet assay (+ Fpg) at 2 h post-treatment in CT, CTA, CTR, CTP, CTRPA and CTRPAt respectively. (**D**) Representative images of 53BP1 (red) immunostaining in CT cell line in control condition or treated with HNE for 2 h. DNA was stained with DAPI (blue). Scale bar = 10 µm (**E**) Quantification of HCECs positives for 53BP1 after 2 h of treatment to contaminants. Cells were scored positive when containing more than five 53BP1 foci. No positive cells were observed in the conditions of untreated CT. All the results (**A**–**C**,**E**) represent the mean ± SEM of three independent experiments; statistical differences were analysed by post-hoc tests of a 2-way ANOVA model between control and treated cells (* *p* < 0.05, ** *p* < 0.01, *** *p* < 0.001) or between CT cells and others isogenic cell lines (# *p* < 0.05, ## *p* < 0.01, ### *p* < 0.001).

**Figure 2 cancers-13-04337-f002:**
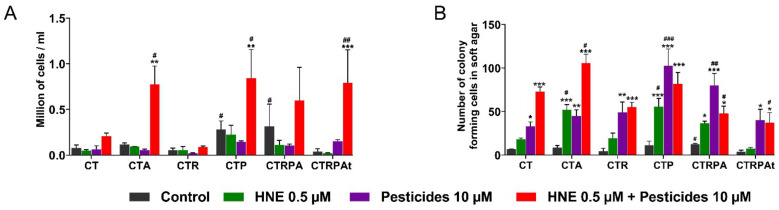
Cell growth and clonogenicity after a three week-exposure to contaminants in HCECs. (**A**) Cell counting at the end of exposure. (**B**) After exposure, HCECs were cultured in soft agar and colonies were quantified after 10 days. Results in (**A**,**B**) panels represent the mean ± SEM of three independent experiments; statistical differences were analysed by post-hoc tests of a 2-way ANOVA model between control and treated cells (* *p* < 0.05, ** *p* < 0.01, *** *p* < 0.001) or between CT cells and others isogenic HCECs (# *p* < 0.05, ## *p* < 0.01, ### *p* < 0.001).

**Figure 3 cancers-13-04337-f003:**
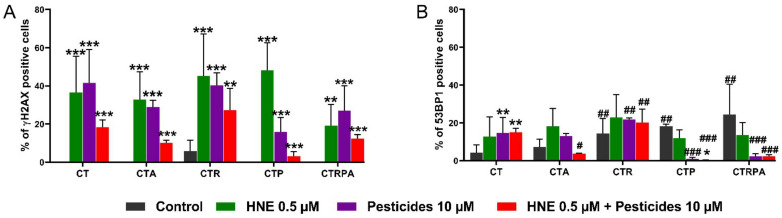
Analysis of DNA damage markers after three weeks exposure in HCECs. (**A**) Quantification of HCECs positives for γH2AX. No positive cells were observed in the conditions of untreated CT, CTA, CTP and CTRPA. (**B**) Quantification of HCECs positives for 53BP1 after a long-term exposure (3 weeks). Results from (**A**,**B**) panels represent the mean ± SEM of three independent experiments; statistical differences were analysed by post-hoc tests of a 2-way ANOVA model between control and treated cells (* *p* < 0.05, ** *p* < 0.01, *** *p* < 0.001) or between CT cells and others isogenic HCECs (# *p* < 0.05, ## *p* < 0.01, ### *p* < 0.001).

**Figure 4 cancers-13-04337-f004:**
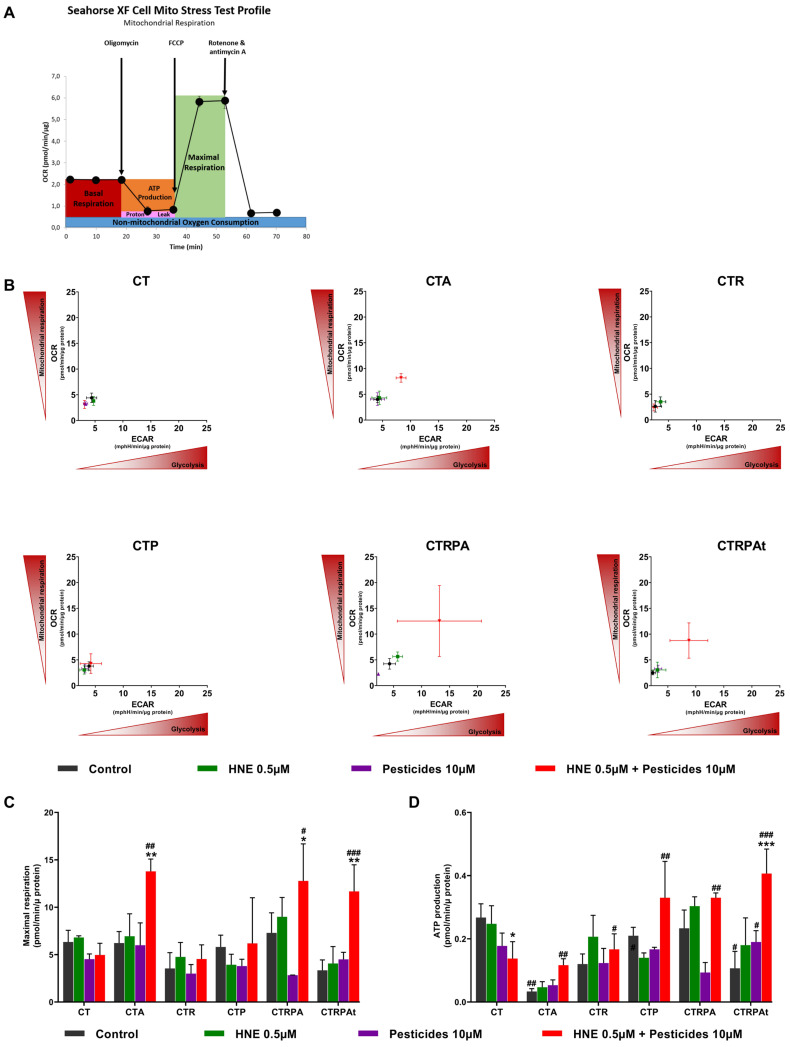
Metabolic reprogramming after three weeks exposure to contaminants in HCECs. (**A**) Agilent Seahorse XF Cell Mito Stress Test profile, showing the key parameters of mitochondrial function. (**B**) Quantification of basal Extracellular Acidification Rate (ECAR) and basal Oxygen Consumption Rate (OCR) using Seahorse^®^ after long term-exposure to contaminants in HCECs. (**C**) Maximal respiratory using Seahorse^®^. (**D**) ATP production using Seahorse^®^. All the results (**B**–**D**) represent the mean ± SEM of three independent experiments; statistical differences were analysed by post-hoc tests of a 2-way ANOVA model between control and treated cells (* *p* < 0.05, ** *p* < 0.01, *** *p* < 0.001) or between HCECs CT cells and others isogenic HCECs (# *p* < 0.05, ## *p* < 0.01, ### *p* < 0.001).

**Figure 5 cancers-13-04337-f005:**
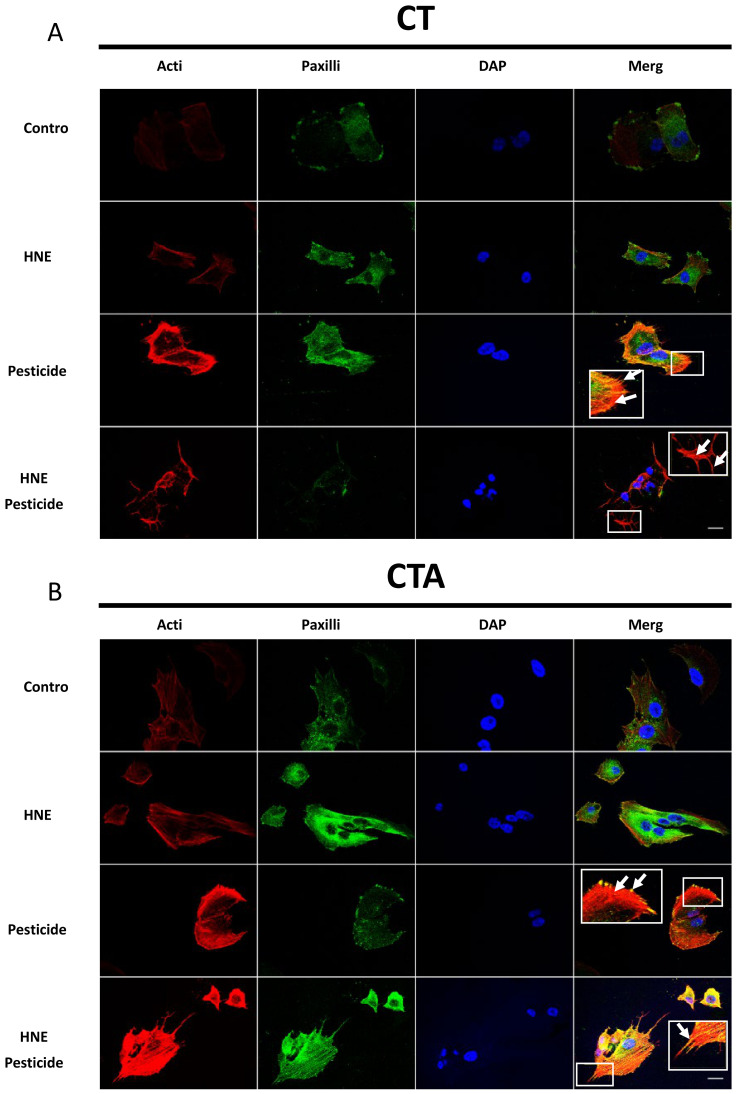
Cell morphology after three weeks exposure to contaminants in HCECs. Representative images of Actin (red), Paxillin (green) immunostaining in the six HCECs, showing the colocalization of Actin and Paxillin (MERGE, yellow). DNA was stained with DAPI (blue). Scale bar = 10 µm. (**A**) CT cells: for pesticide condition, white arrows indicate filipodia, stress fibers and lamellipodia; for HNE-Pest condition white arrows indicate cortical actin and filipodia. (**B**) CTA cells: for pesticide condition, white arrows indicate stress fibers and anchoring points; for HNE-Pest condition arrows indicate filipodia. (**C**) CTR cells, (**D**) CTP cells, (**E**) CTRPA cells and (**F**) CTRPAt cells. (**G**) Summarized table of analysed moving cells parameters after long-term exposure of contaminants.

**Figure 6 cancers-13-04337-f006:**
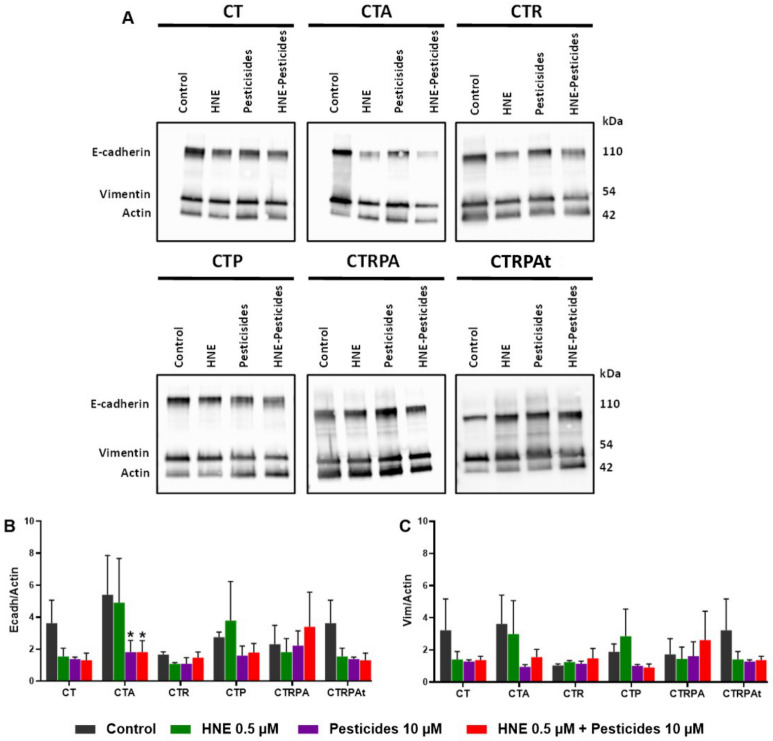
Epithelial-mesenchymal transition (EMT) markers, after three weeks exposure to contaminants in HCECs. (**A**) E-cadherin, Vimentin and Actin immunoblots of soluble extracts. (**B**) Quantification of E-cadherin immunoblots of HCECs relative to actin. (**C**) Quantification of Vimentin immunoblots of HCECs relative to actin. All the results (**B**,**C**) represent the mean ± SEM of three independent experiments; statistical differences were analysed by post-hoc tests of a 2-way ANOVA model between control and treated cells (* *p* < 0.05).

**Figure 7 cancers-13-04337-f007:**
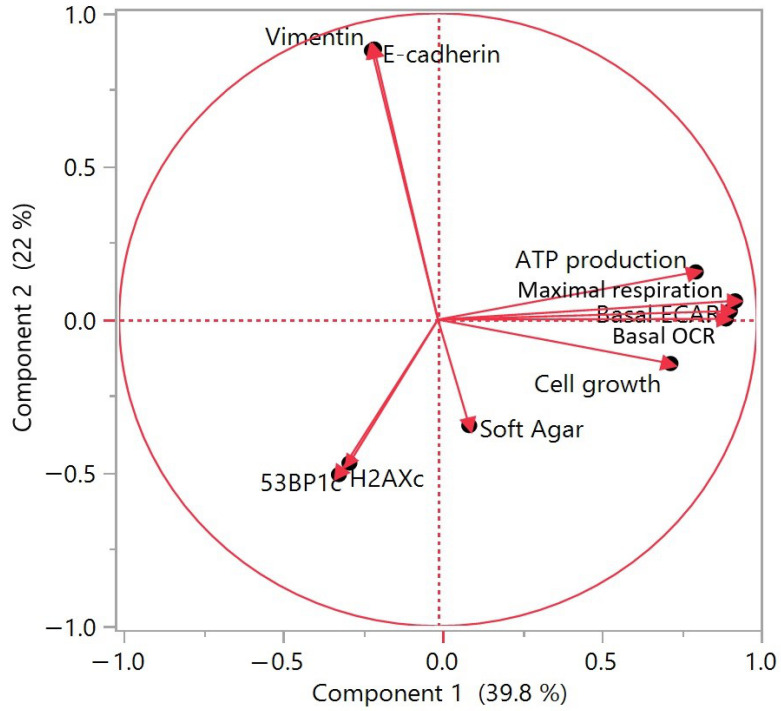
Principal component analysis of long-term exposure results to contaminants in HCECs.

**Table 1 cancers-13-04337-t001:** Composition of mixture of pesticides.

Pesticide	Family	Chemical Family	Proportion	Final Concentration
Procymidone	Fungicide	Dicarboximide	42%	4.2 µM
Iprodione	Fungicide	Dicarboximide	33%	3.3 µM
Cyprodinil	Fungicide	Pyrimidiamine	15%	1.5 µM
Fludioxonil	Fungicide	Phenylpyrrole	9%	0.9 µM
Lambda-Cyhalothrin	Insecticide	Synthetic Pyrethroids	1%	0.1 µM

## Data Availability

Raw data are available upon request.

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
