# Peer review of "Short-Term and Long-Term Carcinogenic Effects of Food Contaminants (4-Hydroxynonenal and Pesticides) on Colorectal Human Cells: Involvement of Genotoxic and Non-Genomic Mechanisms"

_cancers, 2021, doi:10.3390/cancers13174337_

Round 1
Reviewer 1 Report
In this manuscript the Authors analyzed the impact of acute and long-term exposure of two western diet- associated food contaminants at low doses: 4-hydroxynonenal (HNE), and a mixture of pesticides in an in vitro models of normal and genetically modified human colonic cells, exhibiting different genetic susceptibilities to CRC. Showing that a long-term exposure to contaminants induces malignant transformation with different cellular mechanisms, depending on genetic susceptibility and contaminants alone or in mixtures.
Comments:
- explain why CRC is associated with smoke.
- Lane 316 correct Figure 1F with E.
- In Figure 1E in CT lack the control
- In Figure 3A in CT, CTA, CTP AND CTRPA lack the control
Author Response
please see tthe attachment

Reviewer 2 Report
The manuscript ID cancers-1307501 by Arnaud et al., entitled „Short-term and long-term carcinogenic effects of food contaminants (4-hydroxynonenal and pesticides) on colorectal human cells: involvement of genotoxic and non-genomic mechanisms“ represents an interesting addition to the long-term disputes about the aetiology of sporadic colorectal cancer. The possible contribution of food contaminants on carcinogenic process, as tested on cell lines, is interestingly addressed. However, prior to the acceptance the manuscript has to undergo a major revision.
The manuscript is quite extensive. A reduction may help to the readers to comprehend the story better.
The authors should give a clearly spelled rationale for selecting particularly these contaminants. What the authors mean "low concentrations"? Did you re-calculate the dosis for that humans are likely to be exposed? In order to avoid this dispute eliminate please low concentrations.
The authors are kindly asked to review a recent literature. The recent work of Murphy should not be omitted (e.g. Molecular Aspects in Medicine, Gastroenterology etc.). Introduction should be shorter with clearly spellt novelty.
Specific Comments:
L.27: This most frequently occurs in interactions with genetic make-up and microenvironment.
Maybe better to express it: ...to investigate environmental part of colorectal...
L.33: Did the authors consider improper function of DNA repair? For instance, defects in MLH1 function might be of interest.
L.35: Is it possible to be more specific? Any alteration? If moderate, please specify.
L.42: Lipid perodxidation leads to endogenous DNA adducts, such as those of malondialdehyde, etheno-adducts and oxidative DNA damage. The last results in GC:TA transversions, occuring often in KRAS (Vodicka et al. IJMS 2020, review). These aspects have to be discussed more in details.
L.47: Introduction: Is not it so in smokers either? Is it relevant to stratify the patients on smokers and non-smokers for the main message of this paper?
L.60: Please confer the paper of Murphy et al., Mol Asp Med 2019.
L.80: Please mention various interactions of individual components in the mixture.
L.103-105: This sentence is in itself a bit controversial. Is the knowledge accumulated or is there a lack of knowledge? Please specify clearly, what is the novelty or what are the novel aspects.
L.106: Again, look at Murphy et al. MAM 2019.
...interplay of genetic, environmental and microenvironmental factors. The flowchart is shown in Vodicka et al. Pharmacogenomics 2019.
L.110: Not very clearly stated. Please rephrase.
L.205: Does this represent double-strand breaks or homologous repair intermediates?
L.285: This is a bit misleading. If you would do this, than the outcomes would not be comparable. Please re-phrase the sentence.
L.295: Please re-phrase this sentence. It is not clear enough.
L.301: How do you assess DSBs with the method? Fpg-sensitive sites do not reflect 8-OH-dG.
L.363: Or they reflect DNA repair intermediates.
L.461: Age and a length of exposure have to be considered. The composition of microbiota as well. This has to be discussed.
L.505: Did the authors verify the findings in in vivo experiments, at least on mice?
L.546: Did the authors observe any traces of interaction (i.e. addition, potentiation etc.)?
Minor and Editorial Comments:
L.21: What the authors mean by normal? Please specify.
L.37: …later stages. Of what? What was the effect of HNE?
L.79: There is a…
L.177: Table: use consistently decimal dot instead of komma.
L.204: Damage instead of damages.
L.209: ...only as a range...
L.303: irrespective
L.331-332: ...to proliferate and survive. W/O questionmark.
L.334: Here we PRESENT…
L.338 and elsewhere in řthe text: Increase IN something.
L.342: What does the normality mean? Please explain.
L.364: Significantly increased proportion.
L.455: cancer and environment
Round 2
Reviewer 2 Report
The authors have considered all the points risen up and implemented the changes appropriately. The manuscript, which addresses very important aspects of carcinogenesis, is a nice, refreshing story out of the mainstream.